# Variational Inference via
# $\chi$ Upper Bound Minimization

**Adji B. Dieng**
Columbia University

**Dustin Tran**
Columbia University

**Rajesh Ranganath**
Princeton University

**John Paisley**
Columbia University

**David M. Blei**
Columbia University

## Abstract

Variational inference (VI) is widely used as an efficient alternative to Markov chain Monte Carlo. It posits a family of approximating distributions $q$ and finds the closest member to the exact posterior $p$. Closeness is usually measured via a divergence $D(q||p)$ from $q$ to $p$. While successful, this approach also has problems. Notably, it typically leads to underestimation of the posterior variance. In this paper we propose CHIVI, a black-box variational inference algorithm that minimizes $D_\chi(p||q)$, the $\chi$-divergence from $p$ to $q$. CHIVI minimizes an upper bound of the model evidence, which we term the $\chi$ upper bound (CUBO). Minimizing the CUBO leads to improved posterior uncertainty, and it can also be used with the classical VI lower bound (ELBO) to provide a sandwich estimate of the model evidence. We study CHIVI on three models: probit regression, Gaussian process classification, and a Cox process model of basketball plays. When compared to expectation propagation and classical VI, CHIVI produces better error rates and more accurate estimates of posterior variance.

## 1 Introduction

Bayesian analysis provides a foundation for reasoning with probabilistic models. We first set a joint distribution $p(\mathbf{x}, \mathbf{z})$ of latent variables $\mathbf{z}$ and observed variables $\mathbf{x}$. We then analyze data through the posterior, $p(\mathbf{z} \,|\, \mathbf{x})$. In most applications, the posterior is difficult to compute because the marginal likelihood $p(\mathbf{x})$ is intractable. We must use approximate posterior inference methods such as Monte Carlo [1] and variational inference [2]. This paper focuses on variational inference.

Variational inference approximates the posterior using optimization. The idea is to posit a family of approximating distributions and then to find the member of the family that is closest to the posterior. Typically, closeness is defined by the Kullback-Leibler (KL) divergence $\mathrm{KL}(q \,\|\, p)$, where $q(\mathbf{z}; \boldsymbol{\lambda})$ is a variational family indexed by parameters $\boldsymbol{\lambda}$. This approach, which we call KLVI, also provides the evidence lower bound (ELBO), a convenient lower bound of the model evidence $\log p(\mathbf{x})$.

KLVI scales well and is suited to applications that use complex models to analyze large data sets [3]. But it has drawbacks. For one, it tends to favor underdispersed approximations relative to the exact posterior [4, 5]. This produces difficulties with light-tailed posteriors when the variational distribution has heavier tails. For example, KLVI for Gaussian process classification typically uses a Gaussian approximation; this leads to unstable optimization and a poor approximation [6].

One alternative to KLVI is expectation propagation (EP), which enjoys good empirical performance on models with light-tailed posteriors [7, 8]. Procedurally, EP reverses the arguments in the KL divergence and performs local minimizations of $\mathrm{KL}(p \,\|\, q)$; this corresponds to iterative moment matching

on partitions of the data. Relative to KLVI, EP produces overdispersed approximations. But EP also has drawbacks. It is not guaranteed to converge [7, Figure 3.6]; it does not provide an easy estimate of the marginal likelihood; and it does not optimize a well-defined global objective [9].

In this paper we develop a new algorithm for approximate posterior inference, $\chi$-divergence variational inference (CHIVI). CHIVI minimizes the $\chi$-divergence from the posterior to the variational family,

$$D_{\chi^2}(p \,\|\, q) = \mathbb{E}_{q(\mathbf{z};\boldsymbol{\lambda})}\Big[\Big(\frac{p(\mathbf{z} \,|\, \mathbf{x})}{q(\mathbf{z};\boldsymbol{\lambda})}\Big)^2 - 1\Big]. \tag{1}$$

CHIVI enjoys advantages of both EP and KLVI. Like EP, it produces overdispersed approximations; like KLVI, it optimizes a well-defined objective and estimates the model evidence.

As we mentioned, KLVI optimizes a lower bound on the model evidence. The idea behind CHIVI is to optimize an *upper bound*, which we call the $\chi$ upper bound (CUBO). Minimizing the CUBO is equivalent to minimizing the $\chi$-divergence. In providing an upper bound, CHIVI can be used (in concert with KLVI) to sandwich estimate the model evidence. Sandwich estimates are useful for tasks like model selection [10]. Existing work on sandwich estimation relies on MCMC and only evaluates simulated data [11]. We derive a *sandwich theorem* (Section 2) that relates CUBO and ELBO. Section 3 demonstrates sandwich estimation on real data.

Aside from providing an upper bound, there are two additional benefits to CHIVI. First, it is a *black-box inference algorithm* [12] in that it does not need model-specific derivations and it is easy to apply to a wide class of models. It minimizes an upper bound in a principled way using unbiased reparameterization gradients [13, 14] of the exponentiated CUBO.

Second, it is a viable alternative to EP. The $\chi$-divergence enjoys the same "zero-avoiding" behavior of EP, which seeks to place positive mass everywhere, and so CHIVI is useful when the KL divergence is not a good objective (such as for light-tailed posteriors). Unlike EP, CHIVI is guaranteed to converge; provides an easy estimate of the marginal likelihood; and optimizes a well-defined global objective. Section 3 shows that CHIVI outperforms KLVI and EP for Gaussian process classification.

The rest of this paper is organized as follows. Section 2 derives the CUBO, develops CHIVI, and expands on its zero-avoiding property that finds overdispersed posterior approximations. Section 3 applies CHIVI to Bayesian probit regression, Gaussian process classification, and a Cox process model of basketball plays. On Bayesian probit regression and Gaussian process classification, it yielded lower classification error than KLVI and EP. When modeling basketball data with a Cox process, it gave more accurate estimates of posterior variance than KLVI.

**Related work.** The most widely studied variational objective is $\mathrm{KL}(q \,\|\, p)$. The main alternative is EP [15, 7], which locally minimizes $\mathrm{KL}(p \,\|\, q)$. Recent work revisits EP from the perspective of distributed computing [16, 17, 18] and also revisits [19], which studies local minimizations with the general family of $\alpha$-divergences [20, 21]. CHIVI relates to EP and its extensions in that it leads to overdispersed approximations relative to KLVI. However, unlike [19, 20], CHIVI does not rely on tying local factors; it optimizes a well-defined global objective. In this sense, CHIVI relates to the recent work on alternative divergence measures for variational inference [21, 22].

A closely related work is [21]. They perform black-box variational inference using the reverse $\alpha$-divergence $D_\alpha(q \,\|\, p)$, which is a valid divergence when $\alpha > 0$[1]. Their work shows that minimizing $D_\alpha(q \,\|\, p)$ is equivalent to maximizing a lower bound of the model evidence. No positive value of $\alpha$ in $D_\alpha(q \,\|\, p)$ leads to the $\chi$-divergence. Even though taking $\alpha \leq 0$ leads to CUBO, it does not correspond to a valid divergence in $D_\alpha(q \,\|\, p)$. The algorithm in [21] also cannot minimize the upper bound we study in this paper. In this sense, our work complements [21].

An exciting concurrent work by [23] also studies the $\chi$-divergence. Their work focuses on upper bounding the partition function in undirected graphical models. This is a complementary application: Bayesian inference and undirected models both involve an intractable normalizing constant.

## 2  $\chi$-Divergence Variational Inference

We present the $\chi$-divergence for variational inference. We describe some of its properties and develop CHIVI, a black box algorithm that minimizes the $\chi$-divergence for a large class of models.

Variational inference (VI) casts Bayesian inference as optimization [24]. VI posits a family of approximating distributions and finds the closest member to the posterior. In its typical formulation, VI minimizes the Kullback-Leibler divergence from $q(\mathbf{z}; \boldsymbol{\lambda})$ to $p(\mathbf{z} \,|\, \mathbf{x})$. Minimizing the KL divergence is equivalent to maximizing the ELBO, a lower bound to the model evidence $\log p(\mathbf{x})$.

## 2.1 The $\chi$-divergence

Maximizing the ELBO imposes properties on the resulting approximation such as underestimation of the posterior's support [4, 5]. These properties may be undesirable, especially when dealing with light-tailed posteriors such as in Gaussian process classification [6].

We consider the $\chi$-divergence (Equation 1). Minimizing the $\chi$-divergence induces alternative properties on the resulting approximation. (See Appendix 5 for more details on all these properties.) Below we describe a key property which leads to overestimation of the posterior's support.

**Zero-avoiding behavior:** Optimizing the $\chi$-divergence leads to a variational distribution with a *zero-avoiding* behavior, which is similar to EP [25]. Namely, the $\chi$-divergence is infinite whenever $q(\mathbf{z}; \boldsymbol{\lambda}) = 0$ and $p(\mathbf{z} \,|\, \mathbf{x}) > 0$. Thus when minimizing it, setting $p(\mathbf{z} \,|\, \mathbf{x}) > 0$ forces $q(\mathbf{z}; \boldsymbol{\lambda}) > 0$. This means $q$ avoids having zero mass at locations where $p$ has nonzero mass.

The classical objective $\mathrm{KL}(q \,\|\, p)$ leads to approximate posteriors with the opposite behavior, called *zero-forcing*. Namely, $\mathrm{KL}(q \,\|\, p)$ is infinite when $p(\mathbf{z} \,|\, \mathbf{x}) = 0$ and $q(\mathbf{z}; \boldsymbol{\lambda}) > 0$. Therefore the optimal variational distribution $q$ will be $0$ when $p(\mathbf{z} \,|\, \mathbf{x}) = 0$. This *zero-forcing* behavior leads to degenerate solutions during optimization, and is the source of "pruning" often reported in the literature (e.g., [26, 27]). For example, if the approximating family $q$ has heavier tails than the target posterior $p$, the variational distributions must be overconfident enough that the heavier tail does not allocate mass outside the lighter tail's support.[2]

## 2.2 CUBO: the $\chi$ Upper Bound

We derive a tractable objective for variational inference with the $\chi^2$-divergence and also generalize it to the $\chi^n$-divergence for $n > 1$. Consider the optimization problem of minimizing Equation 1. We seek to find a relationship between the $\chi^2$-divergence and $\log p(\mathbf{x})$. Consider

$$\mathbb{E}_{q(\mathbf{z};\boldsymbol{\lambda})}\left[\left(\frac{p(\mathbf{x}, \mathbf{z})}{q(\mathbf{z}; \boldsymbol{\lambda})}\right)^2\right] = 1 + D_{\chi^2}(p(\mathbf{z}|\mathbf{x}) \,\|\, q(\mathbf{z}; \boldsymbol{\lambda})) = p(\mathbf{x})^2[1 + D_{\chi^2}(p(\mathbf{z}|\mathbf{x}) \,\|\, q(\mathbf{z}; \boldsymbol{\lambda}))].$$

Taking logarithms on both sides, we find a relationship analogous to how $\mathrm{KL}(q \,\|\, p)$ relates to the ELBO. Namely, the $\chi^2$-divergence satisfies

$$\frac{1}{2}\log(1 + D_{\chi^2}(p(\mathbf{z}|\mathbf{x}) \,\|\, q(\mathbf{z}; \boldsymbol{\lambda}))) = -\log p(\mathbf{x}) + \frac{1}{2}\log\mathbb{E}_{q(\mathbf{z};\boldsymbol{\lambda})}\left[\left(\frac{p(\mathbf{x}, \mathbf{z})}{q(\mathbf{z}; \boldsymbol{\lambda})}\right)^2\right].$$

By monotonicity of $\log$, and because $\log p(\mathbf{x})$ is constant, minimizing the $\chi^2$-divergence is equivalent to minimizing

$$\mathcal{L}_{\chi^2}(\boldsymbol{\lambda}) = \frac{1}{2}\log\mathbb{E}_{q(\mathbf{z};\boldsymbol{\lambda})}\left[\left(\frac{p(\mathbf{x}, \mathbf{z})}{q(\mathbf{z}; \boldsymbol{\lambda})}\right)^2\right].$$

Furthermore, by nonnegativity of the $\chi^2$-divergence, this quantity is an upper bound to the model evidence. We call this objective the $\chi$ *upper bound (*CUBO*)*.

**A general upper bound.** The derivation extends to upper bound the general $\chi^n$-divergence,

$$\mathcal{L}_{\chi^n}(\boldsymbol{\lambda}) = \frac{1}{n}\log\mathbb{E}_{q(\mathbf{z};\boldsymbol{\lambda})}\left[\left(\frac{p(\mathbf{x}, \mathbf{z})}{q(\mathbf{z}; \boldsymbol{\lambda})}\right)^n\right] = \mathrm{CUBO}_n. \tag{2}$$

This produces a family of bounds. When $n < 1$, $\mathrm{CUBO}_n$ is a lower bound, and minimizing it for these values of $n$ does not minimize the $\chi$-divergence (rather, when $n < 1$, we recover the reverse $\alpha$-divergence and the VR-bound [21]). When $n = 1$, the bound is tight where $\mathrm{CUBO}_1 = \log p(\mathbf{x})$. For $n \geq 1$, $\mathrm{CUBO}_n$ is an upper bound to the model evidence. In this paper we focus on $n = 2$. Other

values of $n$ are possible depending on the application and dataset. We chose $n = 2$ because it is the most standard, and is equivalent to finding the optimal proposal in importance sampling. See Appendix 4 for more details.

**Sandwiching the model evidence.** Equation 2 has practical value. We can minimize the $\mathrm{CUBO}_n$ and maximize the ELBO. This produces a sandwich on the model evidence. (See Appendix 8 for a simulated illustration.) The following *sandwich theorem* states that the gap induced by $\mathrm{CUBO}_n$ and ELBO increases with $n$. This suggests that letting $n$ as close to 1 as possible enables approximating $\log p(\mathbf{x})$ with higher precision. When we further decrease $n$ to 0, $\mathrm{CUBO}_n$ becomes a lower bound and tends to the ELBO.

**Theorem 1 (Sandwich Theorem)**: *Define* $\mathrm{CUBO}_n$ *as in Equation* 2. *Then the following holds:*

- $\forall n \geq 1 \; \mathrm{ELBO} \leq \log p(\mathbf{x}) \leq \mathrm{CUBO}_n.$

- $\forall n \geq 1 \; \mathrm{CUBO}_n$ *is a non-decreasing function of the order $n$ of the $\chi$-divergence.*

- $\lim_{n \to 0} \mathrm{CUBO}_n = \mathrm{ELBO}.$

See proof in Appendix 1. Theorem 1 can be utilized for estimating $\log p(\mathbf{x})$, which is important for many applications such as the evidence framework [28], where the marginal likelihood is argued to embody an Occam's razor. Model selection based solely on the ELBO is inappropriate because of the possible variation in the tightness of this bound. With an accompanying upper bound, one can perform what we call *maximum entropy model selection* in which each model evidence values are chosen to be that which maximizes the entropy of the resulting distribution on models. We leave this as future work. Theorem 1 can also help estimate Bayes factors [29]. In general, this technique is important as there is little existing work: for example, Ref. [11] proposes an MCMC approach and evaluates simulated data. We illustrate sandwich estimation in Section 3 on UCI datasets.

## 2.3 Optimizing the CUBO

We derived the $\mathrm{CUBO}_n$, a general upper bound to the model evidence that can be used to minimize the $\chi$-divergence. We now develop CHIVI, a black box algorithm that minimizes $\mathrm{CUBO}_n$.

The goal in CHIVI is to minimize the $\mathrm{CUBO}_n$ with respect to variational parameters,

$$\mathrm{CUBO}_n(\boldsymbol{\lambda}) = \frac{1}{n} \log \mathbb{E}_{q(\mathbf{z};\boldsymbol{\lambda})} \left[ \left( \frac{p(\mathbf{x}, \mathbf{z})}{q(\mathbf{z}; \boldsymbol{\lambda})} \right)^n \right].$$

The expectation in the $\mathrm{CUBO}_n$ is usually intractable. Thus we use Monte Carlo to construct an estimate. One approach is to naively perform Monte Carlo on this objective,

$$\mathrm{CUBO}_n(\boldsymbol{\lambda}) \approx \frac{1}{n} \log \frac{1}{S} \sum_{s=1}^{S} \left[ \left( \frac{p(\mathbf{x}, \mathbf{z}^{(s)})}{q(\mathbf{z}^{(s)}; \boldsymbol{\lambda})} \right)^n \right],$$

for $S$ samples $\mathbf{z}^{(1)}, ..., \mathbf{z}^{(S)} \sim q(\mathbf{z}; \boldsymbol{\lambda})$. However, by Jensen's inequality, the $\log$ transform of the expectation implies that this is a biased estimate of $\mathrm{CUBO}_n(\boldsymbol{\lambda})$:

$$\mathbb{E}_q \left[ \frac{1}{n} \log \frac{1}{S} \sum_{s=1}^{S} \left[ \left( \frac{p(\mathbf{x}, \mathbf{z}^{(s)})}{q(\mathbf{z}^{(s)}; \boldsymbol{\lambda})} \right)^n \right] \right] \neq \mathrm{CUBO}_n.$$

In fact this expectation changes during optimization and depends on the sample size $S$. The objective is not guaranteed to be an upper bound if $S$ is not chosen appropriately from the beginning. This problem does not exist for lower bounds because the Monte Carlo approximation is still a lower bound; this is why the approach in [21] works for lower bounds but not for upper bounds. Furthermore, gradients of this biased Monte Carlo objective are also biased.

We propose a way to minimize upper bounds which also can be used for lower bounds. The approach keeps the upper bounding property intact. It does so by minimizing a Monte Carlo approximation of the exponentiated upper bound,

$$\mathbf{L} = \exp\{n \cdot \mathrm{CUBO}_n(\boldsymbol{\lambda})\}.$$

---

**Algorithm 1:** $\chi$-divergence variational inference (CHIVI)

---

**Input**: Data $\mathbf{x}$, Model $p(\mathbf{x}, \mathbf{z})$, Variational family $q(\mathbf{z}; \boldsymbol{\lambda})$.

**Output**: Variational parameters $\boldsymbol{\lambda}$.

Initialize $\boldsymbol{\lambda}$ randomly.

**while** not converged **do**

    Draw $S$ samples $\mathbf{z}^{(1)}, ..., \mathbf{z}^{(S)}$ from $q(\mathbf{z}; \boldsymbol{\lambda})$ and a data subsample $\{x_{i_1}, ..., x_{i_M}\}$.

    Set $\rho_t$ according to a learning rate schedule.

    Set $\log \mathbf{w}^{(s)} = \log p(\mathbf{z}^{(s)}) + \frac{N}{M} \sum_{j=1}^{M} p(\mathbf{x}_{i_j} \mid \mathbf{z}) - \log q(\mathbf{z}^{(s)}; \boldsymbol{\lambda}_t)$, $s \in \{1, ..., S\}$.

    Set $\mathbf{w}^{(s)} = \exp(\log \mathbf{w}^{(s)} - \max_s \log \mathbf{w}^{(s)})$, $s \in \{1, ..., S\}$.

    Update $\boldsymbol{\lambda}_{t+1} = \boldsymbol{\lambda}_t - \frac{(1-n) \cdot \rho_t}{S} \sum_{s=1}^{S} \left[ \left( \mathbf{w}^{(s)} \right)^n \nabla_{\boldsymbol{\lambda}} \log q(\mathbf{z}^{(s)}; \boldsymbol{\lambda}_t) \right]$.

**end**

---

By monotonicity of $\exp$, this objective admits the same optima as $\mathrm{CUBO}_n(\boldsymbol{\lambda})$. Monte Carlo produces an unbiased estimate, and the number of samples only affects the variance of the gradients. We minimize it using reparameterization gradients [13, 14]. These gradients apply to models with differentiable latent variables. Formally, assume we can rewrite the generative process as $\mathbf{z} = g(\boldsymbol{\lambda}, \epsilon)$ where $\epsilon \sim p(\epsilon)$ and for some deterministic function $g$. Then

$$\hat{\mathbf{L}} = \frac{1}{B} \sum_{b=1}^{B} \left( \frac{p(\mathbf{x}, g(\boldsymbol{\lambda}, \epsilon^{(b)}))}{q(g(\boldsymbol{\lambda}, \epsilon^{(b)}); \boldsymbol{\lambda})} \right)^n$$

is an unbiased estimator of $\mathbf{L}$ and its gradient is

$$\nabla_{\boldsymbol{\lambda}} \hat{\mathbf{L}} = \frac{n}{B} \sum_{b=1}^{B} \left( \frac{p(\mathbf{x}, g(\boldsymbol{\lambda}, \epsilon^{(b)}))}{q(g(\boldsymbol{\lambda}, \epsilon^{(b)}); \boldsymbol{\lambda})} \right)^n \nabla_{\boldsymbol{\lambda}} \log \left( \frac{p(\mathbf{x}, g(\boldsymbol{\lambda}, \epsilon^{(b)}))}{q(g(\boldsymbol{\lambda}, \epsilon^{(b)}); \boldsymbol{\lambda})} \right). \tag{3}$$

(See Appendix 7 for a more detailed derivation and also a more general alternative with score function gradients [30].)

Computing Equation 3 requires the full dataset $\mathbf{x}$. We can apply the "average likelihood" technique from EP [18, 31]. Consider data $\{\mathbf{x}_1, \ldots, \mathbf{x}_N\}$ and a subsample $\{\mathbf{x}_{i_1}, ..., \mathbf{x}_{i_M}\}$.. We approximate the full log-likelihood by

$$\log p(\mathbf{x} \mid \mathbf{z}) \approx \frac{N}{M} \sum_{j=1}^{M} \log p(\mathbf{x}_{i_j} \mid \mathbf{z}).$$

Using this proxy to the full dataset we derive CHIVI, an algorithm in which each iteration depends on only a mini-batch of data. CHIVI is a black box algorithm for performing approximate inference with the $\chi^n$-divergence. Algorithm 1 summarizes the procedure. In practice, we subtract the maximum of the logarithm of the importance weights, defined as

$$\log \mathbf{w} = \log p(\mathbf{x}, \mathbf{z}) - \log q(\mathbf{z}; \boldsymbol{\lambda}).$$

to avoid underflow. Stochastic optimization theory still gives us convergence with this approach [32].

## 3 Empirical Study

We developed CHIVI, a black box variational inference algorithm for minimizing the $\chi$-divergence. We now study CHIVI with several models: probit regression, Gaussian process (GP) classification, and Cox processes. With probit regression, we demonstrate the sandwich estimator on real and synthetic data. CHIVI provides a useful tool to estimate the marginal likelihood. We also show that for this model where ELBO is applicable CHIVI works well and yields good test error rates.

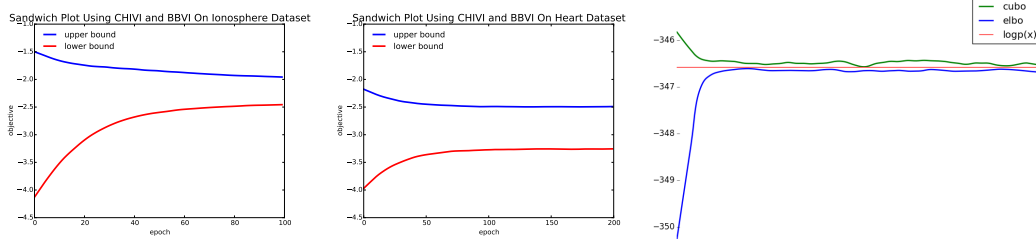

**Figure 1:** Sandwich gap via CHIVI and BBVI on different datasets. The first two plots correspond to sandwich plots for the two UCI datasets *Ionosphere* and *Heart* respectively. The last plot corresponds to a sandwich for generated data where we know the log marginal likelihood of the data. There the gap is tight after only few iterations. More sandwich plots can be found in the appendix.

**Table 1:** Test error for Bayesian probit regression. The lower the better. CHIVI (this paper) yields lower test error rates when compared to BBVI [12], and EP on most datasets.

| Dataset | BBVI | EP | CHIVI |
|---------|------|-----|-------|
| Pima | $0.235 \pm 0.006$ | $0.234 \pm 0.006$ | $\mathbf{0.222 \pm 0.048}$ |
| Ionos | $0.123 \pm 0.008$ | $0.124 \pm 0.008$ | $\mathbf{0.116 \pm 0.05}$ |
| Madelon | $0.457 \pm 0.005$ | $\mathbf{0.445 \pm 0.005}$ | $0.453 \pm 0.029$ |
| Covertype | $0.157 \pm 0.01$ | $0.155 \pm 0.018$ | $\mathbf{0.154 \pm 0.014}$ |

Second, we compare CHIVI to Laplace and EP on GP classification, a model class for which KLVI fails (because the typical chosen variational distribution has heavier tails than the posterior).[3] In these settings, EP has been the method of choice. CHIVI outperforms both of these methods.

Third, we show that CHIVI does not suffer from the posterior support underestimation problem resulting from maximizing the ELBO. For that we analyze Cox processes, a type of spatial point process, to compare profiles of different NBA basketball players. We find CHIVI yields better posterior uncertainty estimates (using HMC as the ground truth).

### 3.1 Bayesian Probit Regression

We analyze inference for Bayesian probit regression. First, we illustrate sandwich estimation on UCI datasets. Figure 1 illustrates the bounds of the log marginal likelihood given by the ELBO and the CUBO. Using both quantities provides a reliable approximation of the model evidence. In addition, these figures show convergence for CHIVI, which EP does not always satisfy.

We also compared the predictive performance of CHIVI, EP, and KLVI. We used a minibatch size of 64 and 2000 iterations for each batch. We computed the average classification error rate and the standard deviation using 50 random splits of the data. We split all the datasets with 90% of the data for training and 10% for testing. For the Covertype dataset, we implemented Bayesian probit regression to discriminate the class 1 against all other classes. Table 1 shows the average error rate for KLVI, EP, and CHIVI. CHIVI performs better for all but one dataset.

### 3.2 Gaussian Process Classification

GP classification is an alternative to probit regression. The posterior is analytically intractable because the likelihood is not conjugate to the prior. Moreover, the posterior tends to be skewed. EP has been the method of choice for approximating the posterior [8]. We choose a factorized Gaussian for the variational distribution $q$ and fit its mean and $\log$ variance parameters.

With UCI benchmark datasets, we compared the predictive performance of CHIVI to EP and Laplace. Table 2 summarizes the results. The error rates for CHIVI correspond to the average of 10 error rates obtained by dividing the data into 10 folds, applying CHIVI to 9 folds to learn the variational parameters and performing prediction on the remainder. The kernel hyperparameters were chosen

**Table 2:** Test error for Gaussian process classification. The lower the better. CHIVI (this paper) yields lower test error rates when compared to Laplace and EP on most datasets.

| Dataset | Laplace | EP | CHIVI |
|---------|---------|-----|-------|
| Crabs | **0.02** | **0.02** | $0.03 \pm 0.03$ |
| Sonar | 0.154 | 0.139 | **0.055 ± 0.035** |
| Ionos | 0.084 | $0.08 \pm 0.04$ | **0.069 ± 0.034** |

**Table 3:** Average $L_1$ error for posterior uncertainty estimates (ground truth from HMC). We find that CHIVI is similar to or better than BBVI at capturing posterior uncertainties. Demarcus Cousins, who plays center, stands out in particular. His shots are concentrated near the basket, so the posterior is uncertain over a large part of the court Figure 2.

| | Curry | Demarcus | Lebron | Duncan |
|------|-------|----------|--------|--------|
| CHIVI | **0.060** | **0.073** | 0.0825 | **0.0849** |
| BBVI | 0.066 | 0.082 | **0.0812** | 0.0871 |

using grid search. The error rates for the other methods correspond to the best results reported in [8] and [34]. On all the datasets CHIVI performs as well or better than EP and Laplace.

### 3.3 Cox Processes

Finally we study Cox processes. They are Poisson processes with stochastic rate functions. They capture dependence between the frequency of points in different regions of a space. We apply Cox processes to model the spatial locations of shots (made and missed) from the 2015-2016 NBA season [35]. The data are from 308 NBA players who took more than $150,000$ shots in total. The $n^{th}$ player's set of $M_n$ shot attempts are $\mathbf{x}_n = \{\mathbf{x}_{n,1}, ..., \mathbf{x}_{n,M_n}\}$, and the location of the $m^{th}$ shot by the $n^{th}$ player in the basketball court is $\mathbf{x}_{n,m} \in [-25, 25] \times [0, 40]$. Let $\mathcal{PP}(\boldsymbol{\lambda})$ denote a Poisson process with intensity function $\boldsymbol{\lambda}$, and $\mathbf{K}$ be a covariance matrix resulting from a kernel applied to every location of the court. The generative process for the $n^{th}$ player's shot is

$$\mathbf{K}_{i,j} = k(\mathbf{x}_i, \mathbf{x}_j) = \sigma^2 \exp(-\frac{1}{2\phi^2}||\mathbf{x}_i - \mathbf{x}_j||^2)$$

$$\mathbf{f} \sim \mathcal{GP}(0, k(\cdot, \cdot)) \; ; \; \boldsymbol{\lambda} = \exp(\mathbf{f}) \; ; \; \mathbf{x}_{n,k} \sim \mathcal{PP}(\boldsymbol{\lambda}) \text{ for } k \in \{1, ..., M_n\}.$$

The kernel of the Gaussian process encodes the spatial correlation between different areas of the basketball court. The model treats the $N$ players as independent. But the kernel $\mathbf{K}$ introduces correlation between the shots attempted by a given player.

Our goal is to infer the intensity functions $\lambda(.)$ for each player. We compare the shooting profiles of different players using these inferred intensity surfaces. The results are shown in Figure 2. The shooting profiles of Demarcus Cousins and Stephen Curry are captured by both BBVI and CHIVI. BBVI has lower posterior uncertainty while CHIVI provides more overdispersed solutions. We plot the profiles for two more players, LeBron James and Tim Duncan, in the appendix.

In Table 3, we compare the posterior uncertainty estimates of CHIVI and BBVI to that of HMC, a computationally expensive Markov chain Monte Carlo procedure that we treat as exact. We use the average $L_1$ distance from HMC as error measure. We do this on four different players: Stephen Curry, Demarcus Cousins, LeBron James, and Tim Duncan. We find that CHIVI is similar or better than BBVI, especially on players like Demarcus Cousins who shoot in a limited part of the court.

## 4 Discussion

We described CHIVI, a black box algorithm that minimizes the $\chi$-divergence by minimizing the CUBO. We motivated CHIVI as a useful alternative to EP. We justified how the approach used in CHIVI enables upper bound minimization contrary to existing $\alpha$-divergence minimization techniques. This enables sandwich estimation using variational inference instead of Markov chain Monte Carlo.

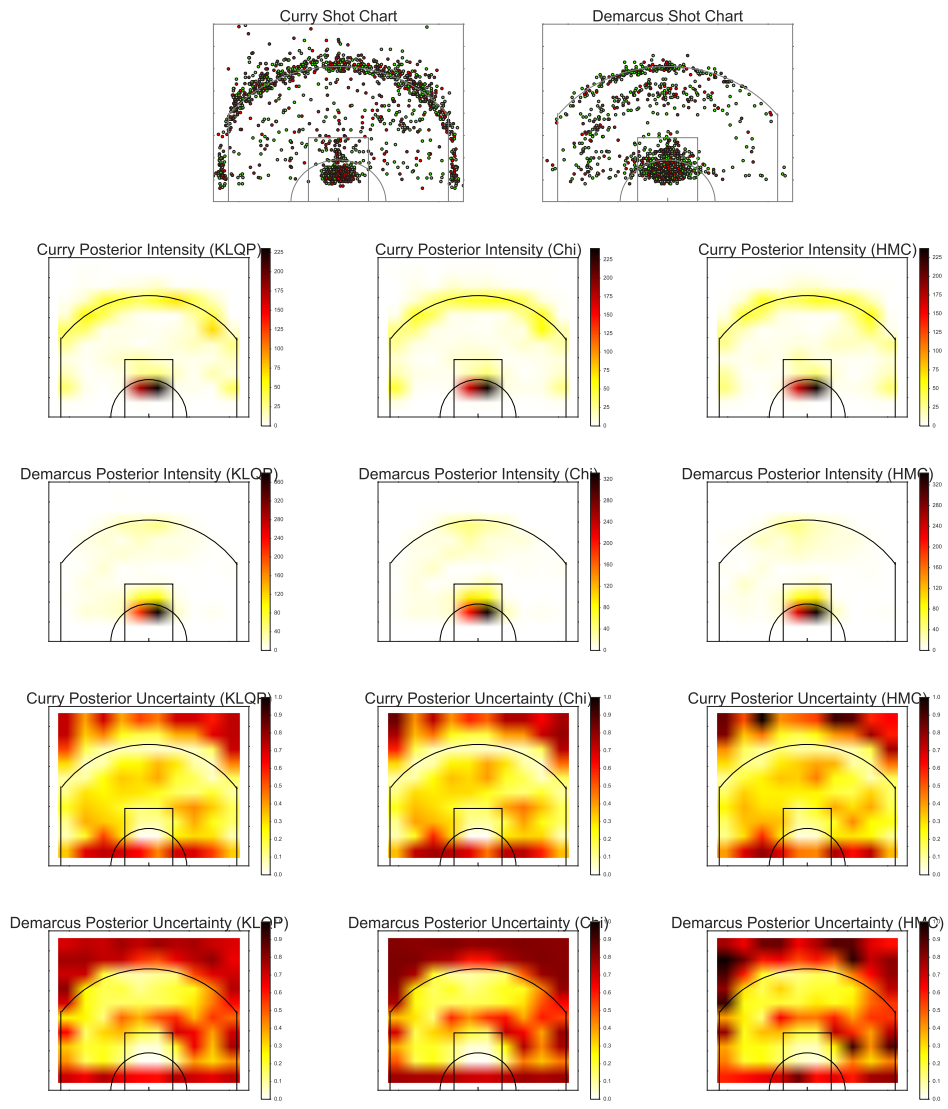

**Figure 2:** Basketball players shooting profiles as inferred by BBVI [12], CHIVI (this paper), and Hamiltonian Monte Carlo (HMC). The top row displays the raw data, consisting of made shots (green) and missed shots (red). The second and third rows display the posterior intensities inferred by BBVI, CHIVI, and HMC for Stephen Curry and Demarcus Cousins respectively. Both BBVI and CHIVI capture the shooting behavior of both players in terms of the posterior mean. The last two rows display the posterior uncertainty inferred by BBVI, CHIVI, and HMC for Stephen Curry and Demarcus Cousins respectively. CHIVI tends to get higher posterior uncertainty for both players in areas where data is scarce compared to BBVI. This illustrates the variance underestimation problem of KLVI, which is not the case for CHIVI. More player profiles with posterior mean and uncertainty estimates can be found in the appendix.

We illustrated this by showing how to use CHIVI in concert with KLVI to sandwich-estimate the model evidence. Finally, we showed that CHIVI is an effective algorithm for Bayesian probit regression, Gaussian process classification, and Cox processes.

Performing VI via upper bound minimization, and hence enabling overdispersed posterior approximations, sandwich estimation, and model selection, comes with a cost. Exponentiating the original CUBO bound leads to high variance during optimization even with reparameterization gradients. Developing variance reduction schemes for these types of objectives (expectations of likelihood ratios) is an open research problem; solutions will benefit this paper and related approaches.

## Acknowledgments

We thank Alp Kucukelbir, Francisco J. R. Ruiz, Christian A. Naesseth, Scott W. Linderman, Maja Rudolph, and Jaan Altosaar for their insightful comments. This work is supported by NSF IIS-1247664, ONR N00014-11-1-0651, DARPA PPAML FA8750-14-2-0009, DARPA SIMPLEX N66001-15-C-4032, the Alfred P. Sloan Foundation, and the John Simon Guggenheim Foundation.

## Footnotes

[1]It satisfies $D(p \,\|\, q) \geq 0$ and $D(p \,\|\, q) = 0 \iff p = q$ almost everywhere

[2]Zero-forcing may be preferable in settings such as multimodal posteriors with unimodal approximations: for predictive tasks, it helps to concentrate on one mode rather than spread mass over all of them [5]. In this paper, we focus on applications with light-tailed posteriors and one to relatively few modes.

[3] For KLVI, we use the black box variational inference (BBVI) version [12] specifically via Edward [33].

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
