[Supplementary Material]

# Variational Inference via $\chi$ Upper Bound Minimization: Supplement

**Adji B. Dieng**
Columbia University
abd2141@columbia.edu

**Dustin Tran**
Columbia University
dustin@cs.columbia.edu

**Rajesh Ranganath**
Princeton University
rajeshr@cs.princeton.edu

**John Paisley**
Columbia University
jpaisley@columbia.edu

**David M. Blei**
Columbia University
david.blei@columbia.edu

## 1 Proof of Sandwich Theorem

We denote by $\mathbf{z}$ the latent variable and $\mathbf{x}$ the data. Assume $\mathbf{z} \in R^D$.

We first show that $\chi$ upper bound $(\text{CUBO})_n$ is a nondecreasing function of the order $n$ of the $\chi$-divergence. Denote by the triplet $(\Omega, \mathcal{F}, Q)$ the probability space induced by the variational distribution $q$ where $\Omega$ is a subspace of $R^D$, $\mathcal{F}$ is the corresponding Borel sigma algebra, and $Q$ is absolutely continuous with respect to the Lebesgue measure $\mu$ and is such that $dQ(\mathbf{z}) = q(\mathbf{z})dz$. Define $w = \frac{p(\mathbf{x},\mathbf{z})}{q(\mathbf{z})}$. We can rewrite $\text{CUBO}_n$ as:

$$\text{CUBO}_n = \frac{1}{n} \log E_q[w^n] = \log\left( (E_q[w^n])^{\frac{1}{n}} \right)$$

Since $\log$ is nondecreasing, it is enough to show
$n \mapsto (E_q[w^n])^{\frac{1}{n}}$ is nondecreasing. This function is the $L_n$ norm in the space defined above:

$$(E_q[w^n])^{\frac{1}{n}} = \left( \int_\Omega |w|^n dQ \right)^{\frac{1}{n}} = \left( \int_\Omega |w|^n q(\mathbf{z})d\mathbf{z} \right)^{\frac{1}{n}}$$

This is a nondecreasing function of $n$ by virtue of the Lyapunov inequality.

We now show the second claim in the sandwich theorem, namely that the limit when $n \to 0$ of $\text{CUBO}_n$ is the evidence lower bound (ELBO). Since $\text{CUBO}_n$ is a monotonic function of $n$ and is bounded from below by ELBO, it admits a limit when $n \to 0$. Call this limit $L$. We show $L = \text{ELBO}$. On the one hand, since $\text{CUBO}_n \geq \text{ELBO}$ for all $n > 0$, we have $L \geq \text{ELBO}$. On the other hand, since $\log t \leq t - 1$; $\forall t > 0$ we have

$$\text{CUBO}_n = \frac{1}{n} \log E_q[w^n] \leq \frac{1}{n}\left[ E_q[w^n] - 1 \right] = E_q\left[ \frac{w^n - 1}{n} \right]$$

$f : n \mapsto w^n$ is differentiable and furthermore
$f'(0) = \lim_{n \to 0}\left[ \frac{w^n - 1}{n} \right] = \log w$. Therefore $\exists n_0 > 0$ such that $|\frac{w^n - 1}{n} - \log w| < 1 \; \forall n < n_0$. Since $||\frac{w^n - 1}{n}| - \log w| < |\frac{w^n - 1}{n} - \log w|$, we have
$|\frac{w^n - 1}{n}| < 1 + \log w$ which is $E_q$-integrable. Therefore by Lebesgue's dominated convergence theorem: $\lim_{n \to 0} E_q\left[ \frac{w^n - 1}{n} \right] = E_q\left[ \lim_{n \to 0} \frac{w^n - 1}{n} \right] = E_q[\log w] = \text{ELBO}$. Since $\text{CUBO}_n$ converges

when $n \to 0$ and $\text{CUBO}_n \leq E_q\left[\frac{w^n - 1}{n}\right] \forall n$, we establish $L \leq \lim_{n \to 0} E_q\left[\frac{w^n - 1}{n}\right] = \text{ELBO}$. The conclusion follows.

## 2 The $\chi$-divergence variational inference (CHIVI) algorithm for small datasets

In the main text we derived a subsampling version of CHIVI. For very small datasets, the average likelihood technique is not needed. The algorithm then uses all the data at each iteration and is summarized in Algorithm 1.

---

**Algorithm 1:** CHIVI without average likelihoods

---

**Input**: Data $\mathbf{x}$, Model $p(\mathbf{x}, \mathbf{z})$, Variational family $q(\mathbf{z}; \boldsymbol{\lambda})$.

**Output**: Variational parameters $\boldsymbol{\lambda}$.

Initialize $\boldsymbol{\lambda}$ randomly.

**while** not converged **do**

> Draw $S$ samples $\mathbf{z}^{(1)}, ..., \mathbf{z}^{(S)}$ from $q(\mathbf{z}; \boldsymbol{\lambda})$.
>
> Set $\rho_t$ from a Robbins-Monro sequence.
>
> Set $\log \mathbf{w}^{(s)} = \log p(\mathbf{x}, \mathbf{z}^{(s)}) - \log q(\mathbf{z}^{(s)}; \boldsymbol{\lambda}_t)$, $s \in \{1, ..., S\}$.
>
> Set $c = \max_s \log \mathbf{w}^{(s)}$.
>
> Set $\mathbf{w}^{(s)} = \exp(\log \mathbf{w}^{(s)} - c)$, $s \in \{1, ..., S\}$.
>
> Update $\boldsymbol{\lambda}_{t+1} = \boldsymbol{\lambda}_t - \frac{(1-n) \cdot \rho_t}{S} \sum_{s=1}^{S} \left[\left(\mathbf{w}^{(s)}\right)^n \nabla_{\boldsymbol{\lambda}} \log q(\mathbf{z}^{(s)}; \boldsymbol{\lambda}_t)\right]$.

**end**

---

## 3 Approximately minimizing an $f$-divergence with CHIVI

In this section we provide a proof that minimizing an $f$-divergence can be done by minimizing a sum of $\chi$-divergencesThese individual $\chi$-divergences can then be optimized via CHIVI. Consider

$$D_f(p \,\|\, q) = \int f\left(\frac{p(x)}{q(x)}\right) q(x) dx$$

Without loss of generality assume $f$ is analytic. The Taylor expansion of $f$ around a given point $x_0$ is

$$f(x) = f(x_0) + f'(x_0)(x - x_0) + \sum_{i=2}^{\infty} f^{(i)}(x_0)\frac{(x - x_0)^i}{i!}$$

Therefore

$$D_f(p \,\|\, q) = f(x_0) + f'(x_0)\left(\mathbb{E}_{q(\mathbf{z} \,|\, \boldsymbol{\lambda})}\left[\frac{p(x)}{q(x)}\right] - x_0\right) + \mathbb{E}_{q(\mathbf{z} \,|\, \boldsymbol{\lambda})}\left[\sum_{i=2}^{\infty} \frac{f^{(i)}(x_0)}{i!}\left(\frac{p(x)}{q(x)} - x_0\right)^i\right]$$

$$= f(x_0) + f'(x_0)(1 - x_0) + \sum_{i=2}^{\infty} \frac{f^{(i)}(1)}{i!}\mathbb{E}_{q(\mathbf{z} \,|\, \boldsymbol{\lambda})}\left[\left(\frac{p(x)}{q(x)} - 1\right)^i\right]$$

where we switch summation and expectation by invoking Fubini's theorem. In particular if we take $x_0 = 1$ the linear terms are zero and we end up with:

$$D_f(p \,\|\, q) = \sum_{i=2}^{\infty} \frac{f^{(i)}(1)}{i!}\mathbb{E}_{q(\mathbf{z} \,|\, \boldsymbol{\lambda})}\left[\left(\frac{p(x)}{q(x)} - 1\right)^i\right] = \sum_{i=2}^{\infty} \frac{f^{(i)}(1)}{i!} D_{\chi^i}(p \,\|\, q)$$

If $f$ is not analytic but $k$ times differentiable for some $k$ then the proof still holds considering the Taylor expansion of $f$ up to the order $k$.

# 4 Importance sampling

In this section we establish the relationship between $\chi^2$-divergence minimization and importance sampling. Consider estimating the marginal likelihood $I$ with importance sampling:

$$I = p(\mathbf{x}) = \int p(\mathbf{x}, \mathbf{z}) d\mathbf{z} = \int \frac{p(\mathbf{x}, \mathbf{z})}{q(\mathbf{z}; \boldsymbol{\lambda})} q(\mathbf{z}; \boldsymbol{\lambda}) d\mathbf{z} = \int w(\mathbf{z}) q(\mathbf{z}; \boldsymbol{\lambda}) d\mathbf{z}$$

The Monte Carlo estimate of $I$ is

$$\hat{I} = \frac{1}{B} \sum_{b=1}^{B} w(\mathbf{z}^{(b)})$$

where $\mathbf{z}^{(1)}, ..., \mathbf{z}^{(B)} \sim q(\mathbf{z}; \boldsymbol{\lambda})$. The variance of $\hat{I}$ is

$$\mathrm{Var}(\hat{I}) = \frac{1}{B} [\mathbb{E}_{q(\mathbf{z}; \boldsymbol{\lambda})} (w(\mathbf{z}^{(b)})^2) - (\mathbb{E}_{q(\mathbf{z}; \boldsymbol{\lambda})} (w(\mathbf{z}^{(b)})))^2] = \frac{1}{B} \left[ \mathbb{E}_{q(\mathbf{z}; \boldsymbol{\lambda})} \left( \left( \frac{p(\mathbf{x}, \mathbf{z}^{(1)})}{q(\mathbf{z}^{(1)}; \boldsymbol{\lambda})} \right)^2 \right) - p(\mathbf{x})^2 \right]$$

Therefore minimizing this variance is equivalent to minimizing the quantity

$$\mathbb{E}_{q(\mathbf{z}; \boldsymbol{\lambda})} \left( \left( \frac{p(\mathbf{x}, \mathbf{z}^{(1)})}{q(\mathbf{z}^{(1)}; \boldsymbol{\lambda})} \right)^2 \right)$$

which is equivalent to minimizing the $\chi^2$-divergence.

# 5 General properties of the $\chi$-divergence

In this section we outline several properties of the $\chi$-divergence.

**Conjugate symmetry** Define

$$f^*(u) = u f(\frac{1}{u})$$

to be the conjugate of $f$. $f^*$ is also convex and satisfies $f^*(1) = 0$. Therefore $D_f^*(p \parallel q)$ is a valid divergence in the $f$-divergence family and:

$$D_f(q \parallel p) = \int f\left(\frac{q(x)}{p(x)}\right) p(x) dx = \int \frac{q(x)}{p(x)} f^*\left(\frac{p(x)}{q(x)}\right) p(x) dx = D_{f^*}(p \parallel q)$$

$D_f(q \parallel p)$ is symmetric if and only if $f = f^*$ which is not the case here. To symmetrize the divergence one can use

$$D(p \parallel q) = D_f(p \parallel q) + D_f^*(p \parallel q)$$

**Invariance under parameter transformation.** Let $y = u(x)$ for some function $u$. Then by Jacobi $p(x)dx = p(y)dy$ and $q(x)dx = q(y)dy$.

$$D_{\chi^n}(p(x) \parallel q(x)) = \int_{x_0}^{x_1} \left(\frac{p(x)}{q(x)}\right)^n q(x) dx - 1 = \int_{y_0}^{y_1} \left(\frac{p(y)\frac{dy}{dx}}{q(y)\frac{dy}{dx}}\right)^n q(y) dy - 1$$

$$= \int_{y_0}^{y_1} \left(\frac{p(y)}{q(y)}\right)^n q(y) dy - 1 = D_{\chi^n}(p(y) \parallel q(y))$$

**Factorization for independent distributions.** Consider taking $p(x, y) = p_1(x) p_2(y)$ and $q(x, y) = q_1(x) q_2(y)$.

$$D_{\chi^n}(p(x, y) \parallel q(x, y)) = \int \frac{p(x, y)^n}{q(x, y)^{n-1}} dx dy = \int \frac{p_1(x)^n p_2(y)^n}{q_1(x)^{n-1} q_2(y)^{n-1}} dx dy$$

$$= \left( \int \frac{p_1(x)^n}{q_1(x)^{n-1}} dx \right) \cdot \left( \int \frac{p_2(y)^n}{q_2(y)^{n-1}} dy \right)$$

$$= D_{\chi^n}(p_1(x) \parallel q_1(x)) \cdot D_{\chi^n}(p_2(y) \parallel q_2(y))$$

Therefore $\chi$-divergence is multiplicative under independent distributions while KL is additive.

**Other properties.** The $\chi$-divergence enjoys some other properties that it shares with all members of the $f$-divergence family namely monotonicity with respect to the distributions and joint convexity.

# 6 Derivation of the CUBO$_n$

In this section we outline the derivation of CUBO$_n$, the upper bound to the marginal likelihood induced by the minimization of the $\chi$-divergence.
By definition:

$$D_{\chi^n}(p(\mathbf{z}\,|\,\mathbf{x})\,\|\,q(\mathbf{z};\boldsymbol{\lambda})) = \mathbb{E}_{q(\mathbf{z};\boldsymbol{\lambda})}\left[\left(\frac{p(\mathbf{z}|\mathbf{x})}{q(\mathbf{z};\boldsymbol{\lambda})}\right)^n - 1\right]$$

Following the derivation of ELBO, we seek an expression of $\log(p(\mathbf{x}))$ involving this divergence. We achieve that as follows:

$$\mathbb{E}_{q(\mathbf{z};\boldsymbol{\lambda})}\left[\left(\frac{p(\mathbf{z}\,|\,\mathbf{x})}{q(\mathbf{z};\boldsymbol{\lambda})}\right)^n\right] = 1 + D_{\chi^n}(p(\mathbf{z}\,|\,\mathbf{x})\,\|\,q(\mathbf{z};\boldsymbol{\lambda}))\mathbb{E}_{q(\mathbf{z};\boldsymbol{\lambda})}\left[\left(\frac{p(\mathbf{x},\mathbf{z})}{q(\mathbf{z};\boldsymbol{\lambda})}\right)^n\right]$$

$$= p(\mathbf{x})^n[1 + D_{\chi^n}(p(\mathbf{z}\,|\,\mathbf{x})\,\|\,q(\mathbf{z};\boldsymbol{\lambda}))]$$

This gives the relationship

$$\log p(\mathbf{x}) = \frac{1}{n}\log\mathbb{E}_{q(\mathbf{z};\boldsymbol{\lambda})}\left[\left(\frac{p(\mathbf{x},\mathbf{z})}{q(\mathbf{z};\boldsymbol{\lambda})}\right)^n\right] - \frac{1}{n}\log(1 + D_{\chi^n}(p(\mathbf{z}\,|\,\mathbf{x})\,\|\,q(\mathbf{z};\boldsymbol{\lambda})))$$

$$\log p(\mathbf{x}) = \text{CUBO}_n - \frac{1}{n}\log(1 + D_{\chi^n}(p(\mathbf{z}|\mathbf{x})\,\|\,q(\mathbf{z};\boldsymbol{\lambda})))$$

By nonnegativity of the divergence this last equation establishes the upper bound:

$$\log p(\mathbf{x}) \leq \text{CUBO}_n$$

# 7 Black Box Inference

In this section we derive the score gradient and the reparameterization gradient for doing black box inference with the $\chi$-divergence.

$$\text{CUBO}_n(\boldsymbol{\lambda}) = \frac{1}{n}\log\mathbb{E}_{q(\mathbf{z};\boldsymbol{\lambda})}\left[\left(\frac{p(\mathbf{x},\mathbf{z})}{q(\mathbf{z};\boldsymbol{\lambda})}\right)^n\right]$$

where $\boldsymbol{\lambda}$ is the set of variational parameters. To minimize $\text{CUBO}_n(\boldsymbol{\lambda})$ with respect to $\boldsymbol{\lambda}$ we need to resort to Monte Carlo. To minimize $\text{CUBO}_n(\boldsymbol{\lambda})$ we consider the equivalent minimization of $\exp\{n \cdot \text{CUBO}(\boldsymbol{\lambda})\}$. This enables unbiased estimation of the noisy gradient used to perform black box inference with the $\chi$-divergence.

**The score gradient** The score gradient of our objective function

$$\mathbf{L} = \exp\{n \cdot \text{CUBO}(\boldsymbol{\lambda})\}$$

is derived below:

$$\nabla_{\boldsymbol{\lambda}}\mathbf{L} = \nabla_{\boldsymbol{\lambda}}\int p(\mathbf{x},\mathbf{z})^n q(\mathbf{z};\boldsymbol{\lambda})^{1-n}d\mathbf{z} = \int p(\mathbf{x},\mathbf{z})^n\nabla_{\boldsymbol{\lambda}}q(\mathbf{z};\boldsymbol{\lambda})^{1-n}d\mathbf{z}$$

$$= \int p(\mathbf{x},\mathbf{z})^n(1-n)q(\mathbf{z};\boldsymbol{\lambda})^{-n}\nabla_{\boldsymbol{\lambda}}q(\mathbf{z};\boldsymbol{\lambda})d\mathbf{z} = (1-n)\int(\frac{p(\mathbf{x},\mathbf{z})}{q(\mathbf{z};\boldsymbol{\lambda})})^n\nabla_{\boldsymbol{\lambda}}q(\mathbf{z};\boldsymbol{\lambda})d\mathbf{z}$$

$$= (1-n)\int(\frac{p(\mathbf{x},\mathbf{z})}{q(\mathbf{z};\boldsymbol{\lambda})})^n\nabla_{\boldsymbol{\lambda}}\log q(\mathbf{z};\boldsymbol{\lambda})q(\mathbf{z};\boldsymbol{\lambda})d\mathbf{z} = (1-n)\mathbb{E}_{q(\mathbf{z};\boldsymbol{\lambda})}\left[\left(\frac{p(\mathbf{x},\mathbf{z})}{q(\mathbf{z};\boldsymbol{\lambda})}\right)^n\nabla_{\boldsymbol{\lambda}}\log q(\mathbf{z};\boldsymbol{\lambda})\right]$$

where we switched differentiation and integration by invoking Lebesgue's dominated convergence theorem. We estimate this gradient with the unbiased estimator:

$$\frac{(1-n)}{B}\sum_{b=1}^{B}\left[\left(\frac{p(\mathbf{x},\mathbf{z}^{(b)})}{q(\mathbf{z}^{(b)};\boldsymbol{\lambda})}\right)^n\nabla_{\boldsymbol{\lambda}}\log q(\mathbf{z}^{(b)};\boldsymbol{\lambda})\right]$$

**Reparameterization gradient** The reparameterization gradient empirically has lower variance than the score gradient. We used it in our experiments. Denote by $L$ the quantity $\exp\{n \cdot \text{CUBO}\}$. Assume $\mathbf{z} = g(\boldsymbol{\lambda},\epsilon)$ where $\epsilon \sim p(\epsilon)$. Then

$$\hat{L} = \frac{1}{B}\sum_{b=1}^{B}\left(\frac{p(\mathbf{x},g(\boldsymbol{\lambda},\epsilon^{(b)}))}{q(g(\boldsymbol{\lambda},\epsilon^{(b)});\boldsymbol{\lambda})}\right)^n$$

is an unbiased estimator of $L$ and its gradient is given by

$$\nabla_{\boldsymbol{\lambda}}\hat{L} = \frac{n}{B}\sum_{b=1}^{B}\left(\frac{p(\mathbf{x}, g(\boldsymbol{\lambda}, \epsilon^{(b)}))}{q(g(\boldsymbol{\lambda}, \epsilon^{(b)}); \boldsymbol{\lambda})}\right)^n \nabla_{\boldsymbol{\lambda}}\log\left(\frac{p(\mathbf{x}, g(\boldsymbol{\lambda}, \epsilon^{(b)}))}{q(g(\boldsymbol{\lambda}, \epsilon^{(b)}); \boldsymbol{\lambda})}\right).$$

# 8 More illustrations

The following figures are results of various experimentations with the CUBO.

**Figure 1**

**Figure 2:** More sandwich plots via CHIVI and black box variational inference (BBVI). The first three plots show simulated sandwich gaps when the order of the $\chi$-divergence is $n = 4$, $n = 2$, and $n = 1.5$ respectively. As we demonstrated theoretically, the gap closes as $n$ decreases. The fourth plot is a sandwich on synthetic data where we know the log marginal likelihood of the data. Here the gap tightens after only 100 iterations. The final two plots are sandwiches on real UCI datasets.

**Figure 3:** More player profiles. Basketball players shooting profiles as inferred by BBVI (**?**), CHIVI (this paper) and Hamiltonian Monte Carlo (HMC). The top row displays the raw data, consisting of made shots (green) and missed shots (red). The second and third rows display the posterior intensities inferred by BBVI, CHIVI and HMC for Lebron James and Tim Duncan respectively. Both BBVI and CHIVI nicely capture the shooting behavior of both players in terms of their posterior mean.The fourth and fifth rows display the posterior uncertainty inferred by BBVI, CHIVI and HMC for Lebron James and Tim Duncan respectively. Here CHIVI and BBVI tend to get similar posterior uncertainty for Lebron James. CHIVI has better uncertainty for Tim Duncan.