[Reviews · NeurIPS 2017]

Reviewer 1



- The paper proposes an upper bound minimization to be added to the LBO of variational inference (VI), resulting in an optimization problem similar in principle to the Squeeze or the sandwich theorem optimizations. - All in all, the idea is interesting and sound. I reckon it would benefit better from comparing to state-of-the-art VI and BBVI algorithms. - The experiments section needs strenghtening, both in terms of the datasets and problems, and (possibly more importantly) in terms of comparing to state-of-the-art VI algorithms. - The variance problem mentioned in the last two lines deserves much more than that, I think. It might be the reason why the method has not been applied to more real-world problems. - Throughout the paper, the note of the fact this algorithm being able to capture advantages of both EP and VI have been thrown a lot. I do not see an enough evidence that the opposite (having problems from both) wouldn't happen at times. Recall that overdispersen in an abstract sense is not a desirable characteristic. It is only desirable when such overdispersion represents the true p with high fidelity. As much as underdispersion is a problem of VI, overdispersion is aslo at many times a problem of EP and an advantage of VI. Maybe a more rigorous study of such cases (at least empricially) would provide further clarification. - Which VI version was used in the first two experiments? Why isn't it one of the recently proposed VI algorithms that have low variance, etc? It would be beneficial to see how the proposed algorithm fares against more refined VI versions. - line 29: "For example, KLVI for Gaussian process classification ...": Is that an example of the phrase right before it? - line 74: "and also revisits [19]": This format of citation is not the best since you're now considering [19] a part of the text without adding up the author' names. - line 198: "probit classification": I reckon it is regression, ain't it? - Refs 21 and 22 belong to the same conference, yet the venue was written differently. More consistence is needed at a NIPS level.

Reviewer 2



Summary The paper presents a chi-divergence based stochastic black-box variational approach (motivated as an alternative to EP), which also enjoys zero-avoiding behavior but with an explicit optimization objective function. As opposed to ELBO based approaches, the proposed method is based on minimizing the chi-divergence upper bound of marginal likelihood, termed as CUBO. The proposed optimization scheme is based on exponentiated CUBO such that the Monte Carlo estimation of the upper bound and gradients are unbiased with preserved upper bound guarantee. Experimental evaluation illustrates sandwich plots on both synthetic and real world datasets, and demonstrate improved test error in classification and posterior uncertainty estimates in Cox process, with comparison against Black-box VI and EP. Qualitative Assessment Variational inference approaches based on maximizing ELBO typically underestimate posterior uncertainty. The chi-divergence based approach proposed in this paper favors over-dispersion by minimizing a stochastic approximation of the upper bound of model evidence. The proposed approach is closely related to the work of Li and Turner, 2016, as is mentioned and differentiated in this paper. The author drew a nice parallel with the relationship between EP (alpha = 1) and KLVI (alpha = 0), and made the flip of arguments as one distinction against [Li and Turner, 2016]. It is interesting to motivate in this way. However, I found it to be a relatively smaller distinction than claimed. Within the alpha-divergence family, the reverse order of p and q is controlled by the parameter alpha or 1-alpha. (1) Setting n = 2 in this paper corresponds to set alpha = -1 in [Li and Turner, 2016], both yield exactly the same form of Chi-upper bound. (2) The flip argument in EP and KLVI leads to completely different optimization algorithms, while here the optimization algorithms are both stochastic variational algorithms, with Monte Carlo approximation, and the “reparametrization trick”. That said, this paper takes a considerably different approach in Monte Carlo approximation. Li and Turner, 2016 uses a naive Monte Carlo approximation on the objective. The estimation is biased. While it works fine for the lower bound when alpha > 0, it becomes troublesome when alpha < 0. First, due to the concavity of logarithm and Jensen inequality, the upper bound guarantee is broken. Second, the bias increases with increasing sample size. This paper avoids these issues by taking the exponential form of the objective and therefore provides unbiased updates. This scheme provides a valuable addition to [Li and Turner, 2016] by being capable of handling upper bound cases in a principled way. But I do have some reservations about whether the exponential form would induce high variance or possible numerical instability as an expense in practice. The sandwich theorem looks very vague and informal. Better specify the range of n in the first two bullets of Theorem 1. The sandwich plots are very interesting. This might be useful for better model selection or hyperparameter optimization in future work. Another suggestion is the presentation of posterior variance estimation results in Section 3.3. The heat maps might be of practical interest, but they do not offer a direct visual clue about whether or not over-or under-estimation occurs and if so, to what extent. Perhaps, a more informative choice would be scatterplots with BBVI, CHIVI, and EP against HMC, respectively. Also, why are the EP results in Table 3 missing? Minor issues: - Line 83, no positive value of alpha … leads to the chi-divergence that we minimizing in this paper. What about negative values? - Line 85: should clarify monotonicity in what? Order? - Line 109-120: the zero-avoiding behavior is described in a mixed context of KL and chi-divergence. Better focused on chi-divergence as the main concern of this paper. - Line 125-126: typo, the last p(z|x) should be p(x, z) - Line 171: to be more clear, consider change "hold" to "exist/matter" Overall, this paper is closely related to the work [Li and Turner, 2016] with a few considerable distinctions made. The proposed chi-divergence method is mainly motivated as an alternative to EP, and the sandwich property of CUBO is illustrated in sandwich plots on both synthetic and real datasets. The exponential objective with unbiased Monte Carlo approximation addresses several issues of [Li and Turner, 2016], and could be useful in handling other upper bound cases as well. There is still some room for improvement in the presentation of results.

Reviewer 3



Continuing on the recent research trend of proposing alternative divergences to perform variational inference, the author proposed a new variational objective CUBO for VI. Inference is equivalent to minimizing CUBO with respect to the variational distributions. CUBO is a proxy to the Chi-divergence between true posteriors and variational distributions, and is also an upper bound of the model's log-evidence. The author claimed that the key advantages of CHIVI over standard VI is the EP-like mass covering property of the resulted variational approximations that tend to over-estimate posterior variance instead of under-estimation as with standard VI. Compared to EP, CHIVI optimizes a global objective function that is guaranteed to converge. The author also claimed that CUBO and ELBO together form a sandwich bound of the marginal likelihood that can be used for model selection. The technical content of the paper appears to be correct albeit some small careless mistakes that I believe are typos instead of technical flaw (see #4 below). The idea of having a sandwich bound for the log-marginal likelihood is certainly good. While the author did demonstrate that the bound does indeed contains the log-marginal likelihood as expected, it is not entirely clear that the sandwich bound will be useful for model selection. This is not demonstrated in the experiment despite being one of the selling point of the paper. It's important to back up this claim using simulated data in experiment. Another key feature of CHIVI is the over-dispersed approximation that it produces, which the author claimed lead to better estimates of the posterior. I believe a more accurate claim would be 'lead to more conservative estimates of the posterior uncertainty', as the goodness of approximation is fundamentally limited by the richness of the variational distributions, which is orthogonal to the choice of the variational objective function. The author provided some evidence for the claim in the Bayesian probit regression, GPC and the basketball player experiments. However, while it is certainly believable that the 'goodness of posterior approximations' plays a role in the test classification error of the first two experiments, the evidence is a little weak because of the discrete nature of the models' outputs. The author can perhaps make a stronger case by demonstrating the superior posterior uncertainty approximation through Bayesian neural network regression, in which the test LL would be more sensitive to the goodness of approximation, or through active learning experiment in which good posterior uncertainty estimates is crucial. There are examples in both https://arxiv.org/pdf/1502.05336.pdf and https://arxiv.org/pdf/1602.02311.pdf. The basketball player example provides pretty good visualisations of the posterior uncertainty, but I'm not entirely sure how to interpret the quantitative results in Table 3 confidently. (e.g., Is the difference of 0.006 between CHIVI and BBVI for 'Curry' a big difference? What is the scale of the numbers?) One aspect that I find lacking in the paper is how computationally demanding CHIVI is compared to standard VI/BBVI? A useful result to present would be side-by-side comparisons of wall clock time required for CHIVI, BBVI and EP in the first two experiments. The variance of the stochastic gradients would certainly play a role in the speed of convergence, and as the author mentioned in the conclusion, the gradients can have high variance. I am quite curious if the reparameterization trick would actually result in lower variance in the stochastic gradients compared to score function gradient when optimizing CUBO? One important use of the ELBO in standard VI is to learn the model's parameters/hyper-parameters by maximizing ELBO wrt the parameters. This appears to be impossible with CUBO as it's an upper bound of the log-marginal likelihood. The author used grid search to find suitable hyper-parameters for the GPC example. However, this is clearly problematic even for a moderately large number of hyper-parameters. How can this be addressed under the CUBO framework? What is the relationship between the grid search metric (I assume this is cross-validation error?) and the sandwich bound for the parameters on the grid? While the paper is pretty readable, there is certainly room for improvements in the clarity of the paper. I find paragraphs in section 1 and 2 to be repetitive. It is clear enough from the Introduction that the key advantages of CHIVI are the zero avoiding approximations and the sandwich bound. I don't find it necessary to be stressing that much more in section 2. Other than that, many equations in the paper do not have numbers. The references to the appendices are also wrong (There is no Appendix D or F). There is an extra period in line 188. The Related Work section is well-written. Good job! Other concerns/problems: 1. In the paper, a general CUBO_n is proposed where n appears to be a hyper-parameter of the bound. What was the choice of n in the experiments? How was it selected? How sensitive are the results to the choice of n? What property should one expect in the resulted approximation given a specific n? Is there any general strategy that one should adopt to select the 'right' n? 2. In line 60, what exactly is an 'inclusive divergence'? 3. In line 118 and 120, the author used the word 'support' to describe the spread of the distributions. I think a more suitable word here would be 'uncertainty'. 4. The equation in line 125 appears to be wrong. Shouldn't there be a line break before the last equal sign, and shouldn't the last expression be equal to E_q[(\frac{p(z,x)}{q(z)})^2]? 5. Line 155: 'call' instead of 'c all' 6. Line 189: The proposed data sub-sampling scheme is also similar to the scheme first proposed in the Stochastic Variational Inference paper by Hoffman et. al., and should be cited? (http://jmlr.org/papers/volume14/hoffman13a/hoffman13a.pdf) 7. Line 203: Does KLVI really fail? Or does it just produce a worse approximations compared to Laplace/EP? Also , 'typically' instead of 'typical'. 8. Line 28/29: KLVI faces difficulties with light-tailed posteriors when the variational distribution has heavier tails. Does CHIVI not face the same difficulty given the same heavy-tailed variational distributions?